# REMEMBER BEFORE YOU EXPLORE: PERSISTENT SHARED MEMORY FOR ZERO-SHOT OBJECT NAVIGATION

## ABSTRACT

In practical applications like home robotics, a single agent over a long lifespan or a team of collaborating agents must perform a continuous stream of tasks in the same environment. However, conventional *zero-shot object navigation* (ZSON) paradigms, which reset memory after each task, are inherently non-collaborative and inefficient for such long-term operations as they lead to redundant exploration. To bridge this gap, we introduce a *Persistent Shared Memory (PSM)* mechanism that allows single or multi-agent systems to accumulate and reuse semantic knowledge across tasks and agents. Our approach builds an *Temporally Consistent Semantic Map (TCSM)*, decoupling scene memory from task-specific information and maintaining semantic consistency via weighted confidence updates. On top of this memory, we design a *beyond-line-of-sight (BLOS) navigation strategy* that propagates stored semantics into nearby navigable areas and performs line-of-sight checks for waypoint selection, enabling reasoning about objects that are currently occluded or distant. Experiments on public benchmarks, including HM3D and MP3D, have shown that our framework avoids redundant scene re-exploration and achieves state-of-the-art performance. Our code will be made available upon acceptance.

## 1 INTRODUCTION

Object navigation is a task in which embodied agents follow natural language instructions to locate a specified goal (Cai et al., 2024). Based on this, zero-shot object navigation (ZSON) further removes the need for task-specific training, relying instead on large pre-trained vision-language models to generalize across unseen tasks and environments (Long et al., 2024).

At the same time, the ultimate goal for embodied agents is to operate and navigate persistently within their environment, like long-lifespan home robots or a collaborative multi-agent team. For instance, an agent dispatched to the bedroom to check a lamp might pass through the kitchen and observe a refrigerator. If a subsequent instruction given to either the same agent or another team member is to navigate to that refrigerator, the system should directly leverage this prior observation instead of re-exploring the house. However, the dominant paradigm in ZSON is fundamentally at odds with this vision. Current approaches (Yokoyama et al., 2024; Huang et al., 2024; Duan et al., 2022) are predominantly stateless, operating with a short-term, task-level memory that is reset after every instruction. This behavior forces the agent into a perpetual cycle of re-exploration, rendering it highly inefficient and unscalable for any practical, long-term deployment.

This motivates us to propose *persistent shared memory (PSM)* mechanism, enabling a single agent to accumulate and share knowledge across tasks and multi-agents in the same environment. Specifically, agents share their observations to a memory space and refer from it to obtain beyond-line-of-sight (BLOS) cues (*i.e.,* cues that beyond the direct visual perception of the current agent) that guide decision making, thereby avoiding redundant scene reexploring and improving navigation efficiency. An example is shown in Fig. 1,

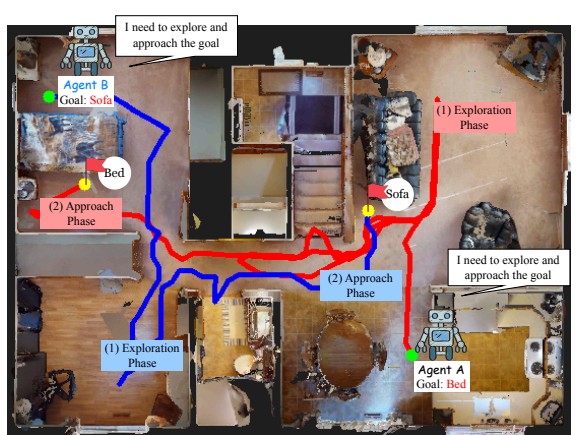 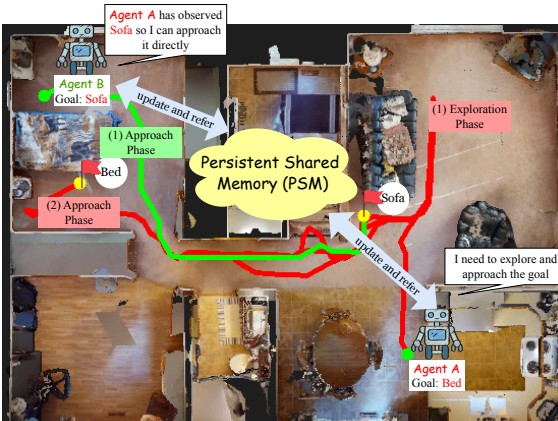

(a) Navigation Without PSM  (b) Navigation With PSM

Figure 1: A later-departing *Agent B* can directly reach the goal without redundant exploration, by leveraging the PSM information previously observed by *Agent A*.

with PSM, the *Agent B* no longer requires unnecessary exploration phase after *Agent A* has already observed its goal during *A*'s navigation task.

However, existing ZSON methods are fundamentally ill-suited for the PSM setting, as their memory mechanisms are either task-entangled or brittle. A significant portion of methods (Yokoyama et al., 2024; Wang & Lee, 2025) rely on implicit value maps tailored to a specific goal. This entanglement of "what the object is" with "how to get there" renders the memory inherently non-transferable and useless for subsequent, different tasks. Other approaches attempt to build explicit semantic maps (Long et al., 2024), but they do so naively. By treating each VLM prediction as ground truth and merging them with greedy updates, they lack any mechanism to handle prediction uncertainty. This brittleness makes them prone to catastrophic error accumulation and semantic drift in a long-term context, where a single misidentification can permanently corrupt the map. Furthermore, the PSM setting introduces a new, unaddressed navigational challenge: leveraging beyond-line-of-sight (BLOS) information. Previous navigation policies are designed only for line-of-sight (LoS) exploration and do not know how to interpret or act on cues from a global map about targets hidden behind walls, leading to inefficient wandering even when the goal's location is known.

To address these challenges, we introduce a novel framework centered around these two key innovations: a robust PSM architecture and a memory-driven navigation policy. First, we propose the *Temporally Consistent Semantic Map (TCSM)*, an explicit confidence-aware 3D voxel map designed for persistence and robustness. At its core, TCSM combats semantic drift and error accumulation through a weighted update mechanism. This approach allows high-confidence, persistent environmental features to be reinforced over time, while effectively filtering out transient noise from segmentation predictions. Second, to leverage this rich memory, we design a *BLOS-aware navigation strategy*. This policy queries the global TCSM to project the location of a known-but-unseen target into the agent's local perception. By evaluating the LoS reachability to these projected goal-hypotheses, the agent can make informed, long-range decisions, planning direct paths towards occluded objects rather than resorting to myopic exploration. Extensive experiments on the HM3D and MP3D benchmarks validate our approach, demonstrating significant gains in navigation efficiency and establishing a new state-of-the-art for persistent ZSON.

Our contributions are as follows: 1) We introduce the **Persistent Shared Memory (PSM)** mechanism for ZSON, reframing the field from isolated, single-task executions towards continuous, collaborative learning and addressing the core inefficiency of stateless navigation; 2) We propose a novel technical framework to realize the PSM setting, featuring two key components: the **Temporally Consistent Semantic Map (TCSM)**

that employs weighted updates to build a robust memory against noise and drift, and a **BLOS-aware navigation strategy** that leverages this persistent knowledge for efficient planning. 3) We conduct extensive experiments on the HM3D (Ramakrishnan et al., 2021) and MP3D (Chang et al., 2017) benchmarks, achieving state-of-the-art performance. Our method significantly improves navigation efficiency (SPL) by eliminating redundant exploration, validating the superiority of our proposed paradigm and framework.

## 2 RELATED WORKS

**Zero-shot object navigation (ZSON)** requires agent to explore and search for a target in the environment through visual observations and the initial text goal. Early works for object navigation build on top of the imitation learning (Silver et al., 2008; Karnan et al., 2022) or reinforcement learning (Kahn et al., 2018; Wöhlke et al., 2021; Cai et al., 2024) in the simulation environments. Since they require large amount of data and annotation for training, leading to challenge to the practical agent deployment. Recently, zero-shot object navigation (Majumdar et al., 2022; Zhou et al., 2023; Wen et al., 2025; Yin et al., 2025), which relies on the off-the-shelf visual perception models (Wu et al., 2024), large language models or vision language models (Radford et al., 2021), without requiring any training and offering strong interpretability. VLFM (Yokoyama et al., 2024) introduce a value map to select frontiers based on the similarity between observation and text goal. GAMap (Huang et al., 2024) uses geometric parts and affordance attributes as the guidance for navigation. InstructNav (Long et al., 2024) proposes to use a dynamic chain-of-navigation and a multi-sourced map to navigate to the goal. g3D-LF (Wang & Lee, 2025) brings pretrained feature fields into VLFM for better scene understanding.

**Long-term shared memory** mechanism is a persistent aspiration for many contemporary intelligent systems, such as SLAM system (Campos et al., 2021). Works like VLMaps (Huang et al., 2023), ConceptGraphs (Gu et al., 2024) and HOV-SG (Werby et al., 2024) try to construct an open-vocabulary representation for 3D scenes. In autonomous driving, the collaborative perception system (Xiong et al., 2023; Xia et al., 2025; Xu et al., 2022) significantly enhances the perception capabilities of individual agents. For instruction-guided navigation, a long-horizon memory systemSong et al. (2025) is required for ongoing decision-making, dynamic re-planning, and sustained reasoning for complex instructions.

Prior ZSON tasks, on one hand, overlooked the crucial aspect of memory sharing across tasks or multiple agents, consequently hindering the sustained utilization of scene information. On the other hand, while approaches like HOV-SG leveraged pre-constructed maps for navigation, they necessitated an initial random walk of the scene by the agent to gather comprehensive scene data and construct semantic maps before any navigation could commence, thereby constraining agent flexibility. Our work addresses these limitations by obviating the need for a preliminary random exploration for map construction prior to task execution within a scene. Concurrently, our method facilitates persistent memory sharing among agents across various tasks and in multi-agent collaborative scenarios, thus catering to more realistic embodied tasks.

## 3 METHODOLOGY

### 3.1 PROBLEM FORMULATION

In the standard ZSON setting, an agent solves an isolated task $T_i = \{g_i, s_i\}$, where it navigates to a goal $g_i$ in an environment $s_i$. The agent's policy $\pi$ relies solely on its history of observations within the current task: $\pi(o_1, \ldots, o_t) \rightarrow a_t$. The memory is transient and the task is self-contained.

To address the inefficiency of this stateless paradigm, we introduce the **Persistent Shared Memory (PSM)** mechanism. This new setting re-frames the problem from solving a single task $T_i$ to solving a **sequence of tasks** $\mathcal{T} = (T_1, T_2, \ldots, T_N)$ within the same environment $s$. Crucially, we introduce a **persistent, shared**

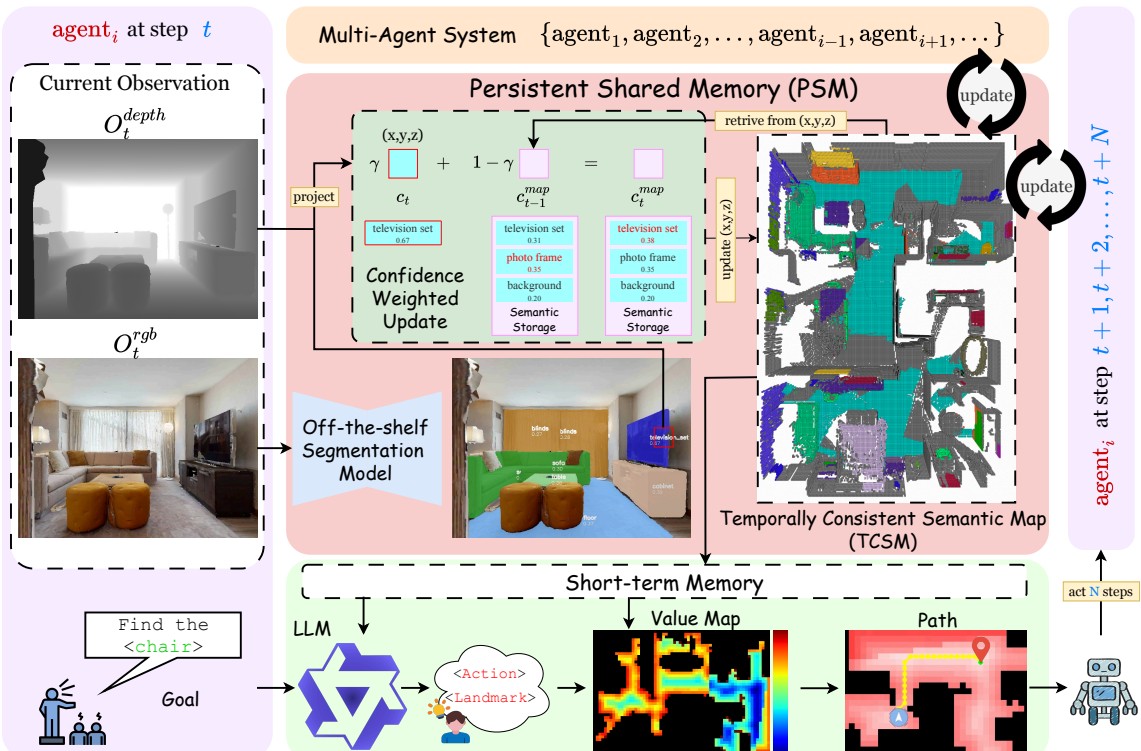

Figure 2: **Overview of our navigation framework.** At each timestep, sensor data $(I_t, D_t)$ is processed by a segmentation model and projected into a 3D voxel map (TCSM). The TCSM maintains a temporally consistent representation by aggregating semantic information over time. For navigation, relevant information is queried from the TCSM to inform the planner, which generates a sequence of executable actions.

**memory state** $M$, which is carried over across all tasks and agents. At each timestep $t$, the agent integrates its new observation $o_t$ into PSM, yielding a transition $M_{t-1} \rightarrow M_t$. The policy then conditions on this updated memory to select the next action: $\pi(o_t, M_t) \rightarrow a_t$. The objective in the PSM setting is thus to leverage the accumulated knowledge in $M$ to improve navigation efficiency over the sequence of tasks.

## 3.2 METHOD OVERVIEW

Our framework provides a concrete realization of the **PSM setting** defined above. This requires answering two central technical questions: (1) How to implement the persistent shared memory state $M$ to be robust and consistent over time? and (2) How to design an effective policy $\pi(o_t, M_t)$ that can leverage this new form of global memory? Our solution, illustrated in Figure 2, is composed of two synergistic components.

First, to implement the memory state $M$, we propose the **Temporally Consistent Semantic Map (TCSM)**. Directly addressing the "brittleness" of naive mapping approaches highlighted in the introduction, the TCSM employs a principled weighted update mechanism. This allows the map to accumulate evidence over time, reinforcing high-confidence semantics while filtering out transient noise from VLM predictions, thus ensuring long-term consistency. Second, to design the memory-conditioned policy $\pi$, we introduce a **BLOS-aware navigation strategy**. This strategy is specifically engineered to exploit the global knowledge stored in the TCSM. It enables the agent to reason about and plan efficient paths towards occluded or distant targets, which is a capability fundamentally absent in traditional, myopic ZSON agents.

The following sections will detail the architecture and update protocols of the TCSM and the mechanics of our integrated navigation policy.

### 3.3 TEMPORALLY CONSISTENT SEMANTIC MAP

The Temporally Consistent Semantic Map (TCSM) is the central component bridging perception and planning. It serves as the concrete implementation of the PSM from our problem formulation.

**Map Representation** We define the map state at time $t$ as a set of tuples $\mathcal{M}_t = \{(v_i, \mathcal{D}_i)\}_{i=1}^{N_t}$, where each tuple represents a known voxel and its associated semantic dictionary. Here, $v_i \in \mathbb{R}^3$ is the coordinate of the $i$-th voxel's center, $N_t$ is the total number of voxels discovered up to time $t$, and $\mathcal{D}_i = \{(s_k, c_k)\}_{k=1}^{K_i}$ is the semantic dictionary for that voxel, containing $K_i$ class-confidence pairs. $\mathcal{M}_t$ supports both spatial and semantic queries. We implement it as a sparse voxel grid, which ensures memory scales only with the explored volume and enables highly efficient lookups suitable for real-time planning.

**Map Update as a State Transition** The map evolution is a state transition process driven by new observations. At each step $t$, the new map state $\mathcal{M}_t$ is generated from the previous state $\mathcal{M}_{t-1}$ and the current observation $o_t = (I_t, D_t)$ with agent pose $P_t$.

The first step is to process the observation $o_t$ to get a set of new semantic information. We back-project each pixel from the current frame into the world, obtaining a set of 3D points. These points are then voxelized, and for each affected voxel, we aggregate the semantic information from all pixels falling into it. This yields a set of new observations, $\mathcal{O}_t = \{(v, s_t^{pix}, c_t^{pix})\}$, where $v$ is the voxel coordinate corresponding to pixel $(i, j)$, and $(s_t, c_t)$ are the semantic and confidence from the perception model. Then the state transition of the map $\mathcal{M}$ can then be formally described as:

$$\mathcal{M}_t = \{(v, \mathcal{D}_v) \in \mathcal{M}_{t-1} \mid v \notin \mathcal{O}_t\} \quad \cup \quad \{(v, \text{Update}(\mathcal{D}_{v,t-1}, v, \mathcal{O}_t)) \mid v \in \mathcal{O}_t\}. \tag{1}$$

This formula elegantly states that the new map $\mathcal{M}_t$ is composed of two parts: 1) The set of all voxels from the old map that were *not* observed in the current step (their state is carried over unchanged). 2) The set of *updated* voxels that were observed in the current step.

The core of the update logic resides in the $\text{Update}(\cdot)$ function. As show in the Fig. 2, for each individual pixel observation $(v, s_t^{pix}, c_t^{pix})$ from $\mathcal{O}_t$ that corresponds to the current voxel $(v_i, \mathcal{D}_{i,t-1})$ in $\mathcal{M}_{t-1}$, the confidence for the class $s_k^{map}$ is updated in-place using a conditional rule:

$$c_{k,t}^{map} = \begin{cases} (1 - \gamma) \cdot c_{k,t-1}^{map} + \gamma \cdot c_t^{pix} & \text{if } s_t^{pix} = s_k^{map} \\ c_t^{pix} & \text{if } s_t^{pix} \notin \{s_k^{map}\}_{k=1}^{K_i} \end{cases} \tag{2}$$

where $c_t^{pix}$ is the confidence from the current pixel observation, and $c_{k,t-1}^{map}$ is the confidence of the class in the dictionary $\mathcal{D}_{i,t-1}$ just before update. This process is repeated for all pixel observations corresponding to voxel $v_i$ within the current timestep $t$.

The definitive semantic label for the voxel is then the class with the highest updated confidence: $s^* = \arg\max_s c_{v,t}(s)$. This formulation precisely defines the map's update mechanism, robustly handling the addition of new voxels and the lifecycle of semantic information within them.

For practical usage, our implementation of the TCSM as a sparse voxel grid ensures that its memory footprint scales only with the *explored* volume, not the entire environment. As the agent navigates and gathers more observations, the map's semantic accuracy and robustness continually improve due to the weighted update mechanism, which reinforces consistent semantics while suppressing transient noise. Querying the map is a direct spatial lookup: given a 3D coordinate, the corresponding voxel's semantic dictionary is retrieved. This operation is highly efficient, making the TCSM suitable for real-time planning decisions.

### 3.4 BLOS-AWARE NAVIGATION STRATEGY

Our navigation planning unfolds in a structured sequence. First, an initial 360° observation sweep updates the TCSM with the latest environmental context. The TCSM is then converted into a short-term memory, which the navigation strategy subsequently utilizes to plan an optimal path to a target waypoint.

**Short-term memory** As show in the Fig. 2, to make the global and task-agnostic TCSM actionable for planning, we transform its information into a short-term memory composed of six specialized point clouds. First, we process the TCSM to derive the environment's geometric structure: navigable spaces ($PC_{nav}$) identified through height filtering and connectivity analysis, the remaining obstacles ($PC_{obs}$), and crucial wall structures ($PC_{wall}$) used for verifying line-of-sight in BLOS navigation. This geometric understanding is complemented by a semantic point cloud ($PC_{sem}$) containing object locations extracted directly from the TCSM. Finally, two dynamic point clouds are utilized: a trajectory trace ($PC_{traj}$) of the agent's recent path to prevent re-exploration, and frontier points ($PC_{frontier}$) to direct exploration towards unknown areas.

**Base navigation strategy** Following InstructNav (Long et al., 2024), we employ Dynamic Chain-of-Navigation (DCoN) with a multi-sourced value map for trajectory planning. DCoN uses a large language model (LLM) to predict a navigation action $\langle Action \rangle$ from $\{Explore, Approach\}$ and a landmark $\langle Landmark \rangle$, based on the task goal and the observed semantic classes from $PC_{sem}$. $PC'_{sem}$ is then filtered from $PC_{sem}$ acrooding to the $\langle Landmark \rangle$. The $\langle Action \rangle$ and $\langle Landmark \rangle$ determine whether the agent should engage in exploration or proceed toward a specific objective. Next, given the task goal and the DCoN output, a vision-language-model (VLM) identifies the next direction for the agent, and the point cloud in that direction is designated as $PC_{intu}$. After DCoN and intuition reasoning, a multi-sourced value map is constructed for trajectory planning. For each source $X$, the value map is based on the minimum distance between points in $PC_{nav}$ and a target point cloud $PC_X \in \{PC'_{sem}, PC_{frontier}, PC_{traj}, PC_{intu}\}$. The distance map $d_X$ is calculated as follows:

$$d_X = \{(p_i, \min_{q_j \in PC_X} \|p_i - q_j\|) | p_i \in PC_{nav}\} \tag{3}$$

The $d_X$ is then normalized into $\hat{d}_X$ using min-max normalization. Once the distance maps for each source are obtained, the final value map $V_{final}^{base}$ is computed by combining the four value maps:

$$V_{sem}^{base} = (1 - \hat{d}_{sem}) \cdot c_{sem}, V_{frontier}^{base} = 1 - \hat{d}_{frontier}, V_{traj}^{base} = \hat{d}_{traj}, V_{intu}^{base} = 1 - \hat{d}_{intu}, \tag{4}$$

$$V_{final}^{base} = (V_{sem}^{base} + V_{frontier}^{base} + V_{traj}^{base} + V_{intu}^{base}) \cdot (d_{obs} > th_{obs}). \tag{5}$$

$c_{sem}$ represents a point cloud composed of the confidence values of the semantic points in $PC'_{sem}$ closest to each navigation point in $PC_{nav}$, which is used to guide the agent to the most reliable semantic goal. Finally, the navigation waypoint for the current planning stage is set to the point with the highest value in $V_{final}^{base}$. The planned trajectory is obtained using the $A^*$ algorithm (Hart et al., 1968).

**BLOS Navigation Strategy** The base navigation strategy is effective for targets located within a contiguous and navigable area. However, it falters when a target is visible (or known via TCSM) but resides in a physically disconnected region. This leads to a state where the agent identifies a target but cannot find a viable path, resulting in inefficient or stalled navigation.

To address this challenge, we introduce a semantic broadcasting mechanism that dynamically adapts the value map for Beyond-Line-of-Sight (BLOS) scenarios. The core principle is to shift the immediate objective from the unreachable target to a strategically chosen "proxy goal region" within the agent's current navigable space. This proxy region is selected based on its clear line-of-sight (LoS) to the final target, guiding the agent to a more advantageous position for subsequent planning. Intuitively, we treat any navigable position that has a potential line-of-sight to the target region as an intermediate proxy goal: once the agent reaches such a position, new observations are likely to expose a feasible path to the target. Concretely, our BLOS procedure first identifies a line-of-sight proxy goal region within the current navigable space and then broadcasts the target semantics over this region to build the value map used for planning.

*Line-of-Sight Region Computation.* To prevent pathing towards obstructions, we first identify a subset of navigable points, $PC'_{nav} \subseteq PC_{nav}$, that have an unobstructed LoS to the target:

$$PC'_{nav} = \{p_i \in PC_{nav} \mid \exists q_j \in PC'_{sem} \text{ s.t. } LoS(p_i, q_j) \text{ is clear}\} \tag{6}$$

Here, the $LoS(p_i, q_j)$ function checks for intersections between the straight-line path from $p_i$ to $q_j$ and the wall point cloud $PC_{wall}$. By construction, $PC'_{nav}$ excludes locations that are behind obstacles with respect to the target, and thus forms a proxy goal region where observing the target becomes geometrically plausible.

*Broadcasted Value Map Construction.* We then compute a new distance map, $d_{blos}$, which measures the distance from every point in the proxy goal region $PC'_{nav}$ to the $PC_{sem}$:

$$d_{blos} = \{(p_i, \min_{q_j \in PC_{sem}} \|p_i - q_j\|) \mid p_i \in PC'_{nav}\} \tag{7}$$

After min-max normalization of $d_{blos}$ to $\hat{d}_{blos}$, we derive the BLOS semantic value map

$$V_{sem}^{BLOS} = (1 - \hat{d}_{blos}) \cdot c_{sem}. \tag{8}$$

Through $V_{sem}^{BLOS}$, points in $PC'_{nav}$ that are closer to the high confidence target receive higher values, creating a smooth potential field that "broadcasts" the target semantics into the currently reachable space and highlights vantage points that are most informative for future planning.

*Final Value Map Fusion.* In BLOS mode, the original semantic component $V_{sem}^{base}$ is replaced by $V_{sem}^{BLOS}$ to produce the final navigation value map, $V_{BLOS}^{final}$:

$$V_{BLOS}^{final} = (V_{sem}^{BLOS} + V_{frontier}^{base} + V_{traj}^{base} + V_{intu}^{base}) \cdot (d_{obs} > th_{obs}) \tag{9}$$

This re-formulation shifts the agent's short-term goal to first navigating to an intermediate location with a better vantage point. Upon reaching this area and acquiring new observations, the target may become directly reachable, at which point the system reverts to the base navigation strategy for the final approach. During the approach phase, only $V_{sem}^{BLOS}$ is taken effect as the goal is clear observed.

## 4 EXPERIMENTS

### 4.1 BENCHMARKS AND IMPLEMENTATION DETAILS

**Benchmarks** We evaluate our approach on two standard benchmarks: HM3D (Ramakrishnan et al., 2021) and MP3D (Chang et al., 2017), all based on the Habitat Simulator (Savva et al., 2019). HM3D include 2000 validation episodes across 20 indoor environments with 6 object goal categories. MP3D contains 2195 validation episodes in 11 indoor environments with 21 object goal categories.

**Metrics** Following the previous works, we use Success Rate (SR) and Success weighted by inverse Path Length (SPL). SR measures the proportion of episodes where the agent reaches the target with a preset distance, and SPL also considers the trajectory length compared to the optimal ground truth trajectory. Since the failed tasks also contributes to the SPL, a higher SR inherently leads to a higher rate of SPL. To more objectively delineate the agent's efficiency in successfully reaching its destination, we use $SuccSPL = SPL/SR$ to measure how the executed trajectory matches the optimal trajectory for all successful tasks.

**Implementation Details** The navigation agent is set to observe images with a resolution of $640 \times 480$ at a height of $0.88m$. Qwen-Max and Qwen-VL-Max from Qwen series (Yang et al., 2025; Wang et al., 2024) are used as LLM and VLM. We set $\gamma = 0.2$ and $th_{obs} = 0.25m$. The voxel size of TCSM is 0.05m. GLEE (Wu et al., 2024) is set as the off-the-shelf segmentation model. The low-level planner that generates actions follows the previous work (Long et al., 2024).

| Method | Zero-Shot | HM3D | | | MP3D | | |
|---|---|---|---|---|---|---|---|
| | | SR | SPL | SuccSPL | SR | SPL | SuccSPL |
| Habitat-Web (Ramrakhya et al., 2022) | ✗ | 41.5 | 16.0 | 38.55 | 31.6 | 8.5 | 26.90 |
| SGM (Zhang et al., 2024) | ✗ | 60.2 | 30.8 | 51.16 | 37.7 | 14.7 | 38.99 |
| ZSON (Majumdar et al., 2022) | ✓ | 25.5 | 12.6 | 49.41 | 15.3 | 4.8 | 31.37 |
| VLFM (Yokoyama et al., 2024) | ✓ | 52.5 | 30.4 | 57.90 | 36.4 | 17.5 | 48.08 |
| ESC (Zhou et al., 2023) | ✓ | 39.2 | 22.3 | 56.89 | 28.7 | 14.2 | 49.48 |
| OpenFMNav (Kuang et al., 2024) | ✓ | 52.5 | 24.1 | 45.90 | 37.2 | 15.7 | 42.20 |
| GAMap (Huang et al., 2024) | ✓ | 53.1 | 26.0 | 48.96 | - | - | - |
| SG-Nav (Yin et al., 2024) | ✓ | 54.0 | 24.9 | 46.11 | 40.2 | 16.0 | 39.80 |
| UniGoal (Yin et al., 2025) | ✓ | 54.5 | 25.1 | 46.06 | 41.0 | 16.4 | 40.00 |
| g3D-LF (Wang & Lee, 2025) | ✓ | 55.6 | 31.8 | 57.19 | 39.0 | 18.8 | 48.21 |
| **Ours** | ✓ | **58.3** | **36.6** | **62.77** | **41.2** | **21.3** | **51.70** |

Table 1: Object Navigation results on HM3D and MP3D.

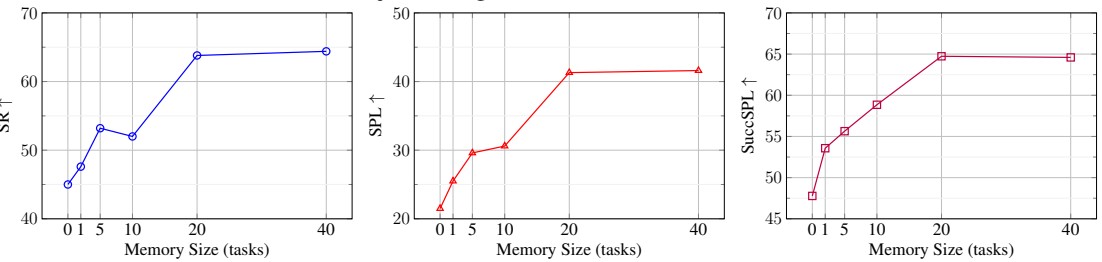

Figure 3: SR, SPL and SuccSPL performance metrics with different size of PSM.

## 4.2 COMPARATIVE ANALYSIS

We evaluate our proposed framework against state-of-the-art object navigation methods, including both learning-based and zero-shot (ZSON) approaches. The quantitative results are summarized in Tab. 1. A key distinction of our method is its ability to share memory across tasks. Unlike baseline methods, which reset their state for each new episode, our agent is equipped with the TCSM. In our experiments, the agent operates sequentially across a series of tasks, allowing the TCSM to persist and grow, thereby simulating a more realistic continual deployment scenario.

The results clearly demonstrate the superiority of our approach. As shown in Tab. 1, our method achieves the highest performance among all ZSON methods on both the HM3D and MP3D benchmarks. On the HM3D benchmark, we outperform g3D-LF (Wang & Lee, 2025) by **+2.7%** in SR and **+4.8%** in SPL. Furthermore, our method surpasses the previous best SuccSPL result from VLFM (Yokoyama et al., 2024) by **+4.87%**. Similarly, our framework consistently achieves the top scores across SR, SPL, and SuccSPL on MP3D.

The substantial improvements, particularly in path-efficiency metrics like SPL and SuccSPL, underscore the core benefit of the TCSM. By retaining and leveraging spatial-semantic knowledge from prior tasks, our agent drastically reduces redundant exploration. While other methods essentially start from scratch in each new scenario, our agent utilizes its accumulated understanding to plan more direct and efficient navigation paths. This capability is crucial for practical applications, as it translates to faster task completion times and more effective long-term operation for autonomous agents in persistent environments.

## 4.3 IMPACT OF PRE-EXISTING MEMORY SIZE

To investigate how navigation performance scales with accumulated knowledge, we designed a controlled experiment on a subset of HM3D dataset with 10 scenes. For each of the 10 scenes, we first generated six memory states by pre-populating a TCSM with observations from $N \in \{0, 1, 5, 10, 20, 40\}$ prior tasks. We then evaluated performance on a held-out set of 500 new tasks. To strictly isolate the benefit of prior

knowledge, the pre-populated TCSM was kept *read-only* during this evaluation phase. The $N = 0$ setting, corresponding to a standard stateless agent, serves as our baseline here.

The results, presented in Fig. 3, show a clear positive correlation between the amount of pre-existing knowledge and task performance. As the number of tasks used to build the TCSM increases, the agent's SR, SPL, and SuccSPL metrics consistently improve. This demonstrates that a more comprehensive initial map allows the agent to reduce exploration, plan more efficient paths, and achieve higher success rates. Furthermore, we observe that the performance gains begin to saturate as the TCSM grows. This suggests that as the memory approaches a complete representation of the environment, the marginal benefit of additional prior knowledge diminishes. The agent's performance gradually converges to an upper bound, which reflects its navigation proficiency in a well-mapped environment.

### 4.4 MULTI-AGENT COLLABORATION WITH TCSM

To evaluate collaborative performance, we simulate $N \in \{1, 2, 4\}$ agents operating concurrently across 10 HM3D scenes. The agents share a single TCSM and execute a total of 80 tasks per scene in synchronized batches of $N$.

| Agent Num | SR | SPL | SuccSPL |
|---|---|---|---|
| 1 agent (w/o PSM) | 52.6 | 25.3 | 48.10 |
| 1 agent | 73.5 (+20.9) | 45.9 (+20.6) | 62.45 |
| 2 agents | 71.1 (+18.5) | 44.1 (+18.8) | 62.03 |
| 4 agents | 70.8 (+18.2) | 43.9 (+18.6) | 62.01 |

Table 2: Performance of multi-agent system.

The shared memory persists across batches, allowing agents to leverage the collective knowledge from all prior executions. We benchmark the multi-agent scenarios ($N = 2, 4$) against both a single agent with persistent memory ($N = 1$) and a standard stateless agent without TCSM (*i.e.*, from scratch) to quantify the benefits of collaboration and memory sharing, respectively.

As presented in Tab. 4.4, all multi-agent configurations leveraging TCSM significantly outperform the from-scratch baseline, confirming the substantial benefits of a shared memory model. However, we observe a slight degradation in average performance metrics as the number of concurrent agents increases. This result stems from the nature of information accumulation in parallel versus sequential execution. In the *single-agent scenario*, each subsequent task directly benefits from the fully completed exploration of all prior tasks. In contrast, in the *4-agent scenario*, the four agents within a given batch start with the same initial memory state. Although they contribute to the TCSM in real-time, they cannot leverage the unexplored information of their parallel peers for their current task. This highlights a trade-off between higher task throughput and the per-task benefit of a more mature, sequentially-enriched memory. Nonetheless, the vast improvement over the from-scratch baseline validates the overall effectiveness of TCSM in multi-agent systems.

### 4.5 ABLATION STUDIES

To demonstrate the effectiveness of our proposed approach, we conduct ablation studies through a progressive integration for submodules. The experiments are conducted on the whole 2000 tasks of HM3D. We also implement a version of InsturctNav with the same LLM and VLM that we deployed in our approach, and extend it with PSM setting to investigate its performance within the PSM setting. Since we employ the same base navigation strategy as InstructNav (Long et al., 2024), our proposed submodules can demonstrate their efficacy when compared to this version.

The ablation results for the submodules are show in Table 3. The implemented version of InstructNav retrive $43.8\%$ of SR and $23.1\%$ of SPL. With PSM setting, it only obtains a gain of $+1.4\%$ in SR and $+1.1\%$ of SPL, due to the deficiencies in its semantic map construction

| Method | PSM Setting | SR | SPL | SuccSPL |
|---|---|---|---|---|
| InstructNav[*] | ✗ | 43.8 | 23.1 | 52.73 |
| InstructNav[*] | ✓ | 45.2 | 24.2 | 53.54 |
| TCSM+Replacement Update+Base Strategy | ✓ | 56.1 | 34.0 | 60.61 |
| TCSM+Weighted Update+Base Strategy | ✓ | **59.1** | 35.2 | 59.56 |
| TCSM+Weighted Update+Full Strategy | ✓ | 58.3 | **36.6** | **62.77** |
| TCSM+Weighted Update+Full Strategy | ✗ | 44.1 | 21.0 | 47.62 |

[*] The re-implemented InstructNav version deploys the same LLM and VLM as ours.

Table 3: Ablation studies on the proposed submodules.

and updating mechanisms. By introducing the proposed TCSM in PSM setting but not using weighted up-

date, a confidence-aware semantic map, the method achieves a gain of $+10.9\%$ in SR, $+9.8\%$ in SPL and $+7.07\%$ in SuccSPL. This version demonstrates comparable efficacy to the majority of preceding methods, while also offering the distinct advantage of TCSM. By further incorporating confidence weighted update, it achieves better SR, but a little drop in SuccSPL. Upon the subsequent integration of our BLOS navigation strategy, we observed a marginal decrease in the method's SR but a notable enhancement in SPL and Succ-SPL. This demonstrates that the BLOS navigation strategy effectively augments an agent's task execution efficiency, particularly in scenarios involving BLOS conditions and disconnected navigable regions.

We also evaluated the performance of TCSM with various segmentation models and different $\gamma$ configurations in Table 4. We sampled four distinct scenarios in HM3D with 396 episodes in total and reported the ablation results above. We deploy XDecoder (Zou et al., 2023), OpenSeeD (Zhang et al., 2023) and GLEE (Wu et al., 2024) as the off-the-shelf segmentation model in PSM. GLEE outperforms the others in Table 4 with higest SR and SPL. Furthermore, the ablation study on $\gamma$ consistently indicated that a setting of 0.8 yielded the most favorable results.

| Segmentation Model | $\gamma$ | SR | SPL |
|---|---|---|---|
| XDecoder (Zou et al., 2023) | 0.2 | 60.1 | 36.6 |
| OpenSeeD (Zhang et al., 2023) | 0.2 | 75.5 | 44.9 |
| GLEE | 0.8 | 82.1 | 50.9 |
| GLEE | 0.6 | 80.1 | 49.3 |
| GLEE | 0.4 | 80.1 | 48.5 |
| GLEE (Wu et al., 2024) | 0.2 | **83.1** | **51.4** |

Table 4: Comparison across segmentation models and $\gamma$ settings.

### 4.6 COMPUTATION COST ANALYSIS

We evaluated the average execution time for TCSM's update and query operations, as well as the system resource consumption of our navigation method, under varying voxel size settings. Experiments were conducted using a system equipped with a 64-core vCPU, 256GB of RAM, and an RTX 3090 GPU with 24GB memory. Results were computed during the 13th task, leveraging the TCSM's accumulated memory from the preceding 12 tasks. The results are shown in Table 5. It was observed that both the update and query times for TCSM, along with the final map file size, exhibit an approximately linear relationship with the increasing number of voxels.

| Voxel Size (m) | 0.02 | 0.05 | 0.08 | 0.10 |
|---|---|---|---|---|
| Voxel Num | 1190k | 188k | 80k | 49k |
| Semantic Update | 434ms | 81ms | 38ms | 22ms |
| Pointcloud Query | 1432ms | 249ms | 92ms | 50ms |
| Semantic Pointcloud Query | 424ms | 90ms | 40ms | 20ms |
| System Memory Usage | 8.86G | 6.30G | 5.91G | 5.72G |
| CUDA Usage | 1.14G | 1.14G | 1.14G | 1.14G |
| Map File Size | 380M | 61M | 26M | 16M |

Table 5: Computation cost analysis.

## 5 CONCLUSION

In this paper, we formulate a persistent shared memory (PSM) mechanism for ZSON task, which enables the agent to share its own memories of the environment while simultaneously reviewing its own or other agents' observations from other tasks. Building on this mechanism, we propose temporally consistent semantic map (TCSM) for task-agnostic scene understanding and cross-task usage. Furthermore, we extend a beyond-line-of-sight (BLOS) navigation strategy to address the challenge of BLOS arising from the introduction of PSM. Based on these developments, our navigation approach achieves the state-of-the-art performance in HM3D and MP3D benchmarks, demonstrating its effectiveness. Our work underscores the critical role of persistent shared memory in navigation tasks, thereby laying a foundational basis for subsequent, more profound long-term collaboration among multi-agent system.

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

## A    LIMITATION

Although our proposed method works well in ZSON, some limitations are considered for future research.

**Dependency on Upstream Perception Models** Our system's perceptual accuracy is capped by the performance of the pre-trained perception models it relies on. While TCSM effectively filters random noise, it remains vulnerable to systematic model biases. A key future challenge is to create a feedback loop where the agent can actively verify or correct its map. This could involve active perception strategies to gather disambiguating evidence or leveraging top-down commonsense knowledge to flag and potentially fix semantic inconsistencies.

**Assumption of a Static Environment** The current implementation of TCSM implicitly assumes a static environment, while this assumption is often violated in real-world, human-centric environments where objects are moved, doors are opened or closed, and layouts change. These dynamics can lead to fake objects persisting in the map or new obstacles going undetected, potentially causing navigation failures. Extending our framework to handle dynamic environments, perhaps by incorporating change detection mechanisms or modeling object permanence and state, is a crucial step towards true long-term autonomy.

## B    STATEMENTS

**Ethics statement** Our research on persistent shared memory for embodied agents aims to create more efficient and capable autonomous systems for beneficial applications, such as assistive robotics in homes or collaborative search-and-rescue operations. However, we acknowledge that technology capable of building detailed, long-term maps of an environment raises significant ethical considerations. Potential risks include misuse for surveillance, which could compromise individual privacy, and the dual-use potential in applications we do not endorse, such as autonomous weaponry. Furthermore, the reliability of such systems is paramount, as errors in memory could lead to unsafe behavior. For the data security, our work is conducted exclusively on publicly available academic datasets (HM3D, MP3D) that contain no personally identifiable information. We advocate for the development of strong privacy-preserving safeguards and clear regulatory guidelines to accompany any real-world deployment of this technology, and we are committed to contributing to a future where embodied AI is developed and used responsibly.

**Reproducibility statement** To ensure reproducibility, we base our experiments on the publicly available HM3D (Ramakrishnan et al., 2021) and MP3D (Chang et al., 2017) datasets within the Habitat simulator. The baseline methods we compare against are also based on publicly available open-source implementations. Our framework utilizes the off-the-shelf model GLEE (Wu et al., 2024) without any fine-tuning. We will release our complete source code, including evaluation scripts and configuration files with all hyperparameters upon publication. The implementation is based on PyTorch and standard libraries for embodied AI research. Detailed instructions for setting up the environment and reproducing our experimental results will be provided with the code release.

**LLM usage statement.** Except using LLM/VLM in our experiments as they are a part of our ZSON approach and baseline (Long et al., 2024), during the preparation of this manuscript, we utilized a large language model (LLM) for the purpose of language editing and proofreading. As non-native English speakers, our goal was to enhance the clarity, grammar, and overall readability of the paper to ensure a better reading experience for the audience. We want to explicitly state that the LLM was not used for any part of the core intellectual work. This includes, but is not limited to, the generation of research ideas, literature review, conceptualization of the methodology, data analysis, or the drawing of conclusions. The authors take full and sole responsibility for all intellectual content, claims, and the final wording of this paper.

