# OpenReview forum: "Remember Before You Explore: Persistent Shared Memory for Zero-Shot Object Navigation"
_ICLR.cc/2026/Conference — ICLR 2026 Conference Withdrawn Submission_

### Official Review · Reviewer_5JYw · 2025-10-27

**Soundness:** 2
**Presentation:** 2
**Contribution:** 2
**Rating:** 4
**Confidence:** 4

**Summary:**

This paper introduces a Persistent Shared Memory (PSM) mechanism for Zero-Shot Object Navigation (ZSON), aiming to improve efficiency in long-term or multi-agent scenarios by reusing environmental knowledge. The framework consists of a Temporally Consistent Semantic Map (TCSM), a 3D voxel grid updated via a confidence-weighted mechanism, and a Beyond-Line-of-Sight (BLOS) navigation strategy that queries this map. The authors claim this approach improves navigation efficiency (SPL) on simulated benchmarks (HM3D, MP3D) by avoiding redundant exploration. The paper is overall clear, technically sound, but its conceptual/practical contribution feels limited. See more in the weakness and questions section.

**Strengths:**

- The paper correctly highlights the significant inefficiency of stateless exploration in ZSON, which is a key bottleneck for practical, long-term embodied AI.
- The high-level idea of a persistent, shared semantic map (PSM) that is robust to some perception noise (TCSM) is a logical and appealing direction for the field.
- The paper is well-structured.

**Weaknesses:**

1. Insufficient Contribution & Lack of Practical Validation: The paper's contribution to the robotics community, which it targets, is undermined by a lack of practical validation. To the ICLR community, its novelty from a learning perspective is limited; it does not introduce a general-purpose learning principle, but rather a specialized application. I would focus to emphasis its to robotics community by following the weakness points:
- Simulation-Only: The entire evaluation is performed in a simulator. For a method claiming to solve a key problem in "home robotics," the absence of any real-world experiments, even a simple demonstration video on a physical robot, is a major omission. Without even small-scale physical validation, the claim of “practical efficiency” is overclaimed and the paper’s contribution is limited.
- Missing Runtime Analysis: The paper claims to improve "efficiency" (SPL), but this metric is about path length, not computational cost. Table 1 is conspicuously missing inference time / runtime analysis. A "persistent" system that is computationally too slow to run in real-time is not a practical solution.
2. Implicit Static Environment Assumption: The method appears to fundamentally assume a static world. The paper does not discuss how the TCSM would handle dynamic objects, changing furniture, or even opening/closing doors. This is a critical limitation that sidesteps one of the hardest challenges of long-term operation in real environments.
3. Parameter explanation missing: The parameter γ (used in Eq. 2) is not explained conceptually, only given as a constant in experiments. Since γ determines the memory update dynamics, its role and sensitivity should be analyzed through an ablation study.
4. Section 3.4 (BLOS strategy) lists procedures but lacks intuitive explanations or reasoning behind design choices. It feels more like code documentation than scientific exposition.
5. Superficial Handling of "Uncertainty": The paper claims to handle "prediction uncertainty" (Line 70), but the proposed solution (Eq. 2) is just a temporal smoothing of confidence scores. The downstream navigation (Sec 3.4) then appears to just take the argmax (highest confidence), rather than truly incorporating the uncertainty distribution into the decision-making policy.
6. Inherited Model Biases: The system's long-term reliability is entirely dependent on the upstream foundation models. The paper does not convincingly address how the map would defend against gradual degradation from systematic biases or errors from these models.

**Questions:**

1. Would it be possible to include a qualitative or real-robot demonstration video to validate the method’s practical relevance?
2. Could you provide a quantitative analysis of the memory (GPU) and computational (update/query time) costs of the TCSM? How do these costs scale with the size of the explored environment and the map resolution?
3. How different is TCSM conceptually from VLMaps or ConceptGraphs, apart from the weighted update rule?
4. Minor Clarifications & Typos:
- In Figure 3, the caption mentions "PSSM." Is this a typo for PSM?
- To better highlight the benefit of persistence in Figure 3, would it be possible to add the performance of a strong stateless baseline? Its performance should be a flat line, which would make the cumulative benefit of your method much clearer.
- Line 28 ("Object navigation challenges embodied agents...") is grammatically confused because it treats object navigation as an active subject that “challenges” the agents. This makes the sentence sound unnatural. A clearer phrasing would be: “Object navigation is a task in which embodied agents follow natural language instructions to locate a specified goal.”
- Please add a brief one-sentence explanation for "beyond-line-of-sight (BLOS) cues" on its first use (Line 43).
- Please fix the non-standard quotation marks (e.g., L267, L280).

I am open to adjusting my evaluation based on the authors' responses during the rebuttal phase.

---

> ### Author Response · Authors · 2025-11-24
> **Response by Authors -- Part 1**
>
> We thank the reviewer for their detailed feedback, which helps us strengthen the paper from a practical robotics perspective.
>
> **Regarding Weaknesses:**
>
> 1.  **Contribution & Practical Validation:**
>     *   **Simulation-Only Evaluation:** We appreciate the reviewer's concern regarding real-world validation. We offer the following clarifications:
>         *   **Standard Benchmarks:** HM3D and MP3D are the standard, widely-accepted benchmarks for Object Navigation. Evaluating on these simulators ensures a fair and direct comparison with prior ZSON methods, which predominantly follow this evaluation protocol. The ZSON task definition itself does not strictly mandate physical robot experiments.
>         *   **Inherent Sim-to-Real Generalization:** Our method is inherently suitable for direct deployment on physical robots. As a zero-shot framework, it relies on large-scale pre-trained vision (e.g., for segmentation) and vision-language models (for planning), combined with a deterministic navigation algorithm. No models were trained or fine-tuned using the simulator data. These foundation models are trained on vast amounts of real-world data, granting them strong generalization capabilities that significantly mitigate the sim-to-real gap. Consequently, we believe the results obtained in HM3D and MP3D are representative and valid.
>         *   **Fairness and Reproducibility:** While physical robot experiments are undoubtedly valuable, they introduce numerous uncontrolled variables (e.g., hardware differences, lighting fluctuations, sensor noise) that can make fair, reproducible, and scientifically rigorous comparisons between different methods challenging. Our simulated setup ensures a controlled environment for equitable evaluation.
>     *   **Missing Runtime Analysis:** We have incorporated an analysis of this aspect into the revised paper (Sec. 4.6), investigating the practical operational cost of TCSM at varying map resolutions by adjusting the voxel size. And it's also shown in below:
>
>         | Voxel Size (m) | 0.02 | 0.05 | 0.08 | 0.10 |
>         | :--- | ---: | ---: | ---: | ---: |
>         | Voxel Num | 1190k | 188k | 80k | 49k |
>         | Semantic Update | 434ms | 81ms | 38ms | 22ms |
>         | Pointcloud Query | 1432ms | 249ms | 92ms | 50ms |
>         | Semantic Pointcloud Query | 424ms | 90ms | 40ms | 20ms |
>         | System Memory Usage | 8.86G | 6.30G | 5.91G | 5.72G |
>         | CUDA Usage | 1.14G | 1.14G | 1.14G | 1.14G |
>         | Map File Size | 380M | 61M | 26M | 16M |
>
> 2.  **Static Environment Assumption:** The standard ZSON benchmarks we use are indeed static. However, our method is not fundamentally limited to static worlds. The TCSM is designed to **continuously update** based on current observations. If an object is moved or a door's state changes, subsequent observations of that area will update the confidence scores in the corresponding voxels, allowing the map to adapt to dynamic changes over time. Investigating long-term navigation in dynamic environments under a ZSON setting is an exciting direction for future work.
>
> 3.  **Parameter γ Explanation:** We apologize for the lack of conceptual explanation for γ. This parameter controls the temporal smoothing in our confidence update rule, balancing the weight of historical evidence against new observations. We have incorporated this ablation study into the revised version (Table 4). And it's also shown in below:
>
>     | Segmentation Model | $\gamma$ | SR | SPL |
>     | :--- | :---: | :---: | :---: |
>     | XDecoder | 0.2 | 60.1 | 36.6 |
>     | OpenSeeD | 0.2 | 75.5 | 44.9 |
>     | GLEE | 0.8 | 82.1 | 50.9 |
>     | GLEE | 0.6 | 80.1 | 49.3 |
>     | GLEE | 0.4 | 80.1 | 48.5 |
>     | GLEE | 0.2 | **83.1** | **51.4** |
>
> 4.  **Clarity of BLOS Strategy Section:** We apologize that the description of the BLOS strategy was unclear. We have restructured this section in the revised paper, clarifying the specific intent of each component.
>
> 5.  **Handling of Uncertainty:** We appreciate the reviewer's call for clarification. Our framework handles uncertainty in two key ways, which we will emphasize more clearly in the revision. First, as the reviewer notes, Eq. 2 provides temporal smoothing of confidence scores. Second, and more importantly, this uncertainty (confidence) is directly incorporated into the downstream navigation policy. During planning, the value assigned to each point in the semantic value map is **weighted by the confidence of its corresponding semantic voxel**. This guides the agent not just toward regions with the correct semantics, but specifically toward regions where the semantic prediction is most *confident*. We sincerely apologize for the omission of this detail in the original manuscript. We have now incorporated these specifics into the revised paper (see Equations 4 and 8).

---

> ### Author Response · Authors · 2025-11-24
> **Response by Authors -- Part 2**
>
> 6.  **Inherited Model Biases:** This is a valid concern for any system relying on foundation models. Like other ZSON methods, our framework's performance is indeed dependent on the generalization capabilities of the pre-trained models. However, these models are trained on vast, diverse, and high-quality real-world datasets, which makes them inherently resistant to systematic degradation from common biases. Furthermore, our TCSM's temporal aggregation mechanism can help mitigate the impact of sporadic model errors, providing an additional layer of robustness.
>
> **Regarding Questions:**
>
> 1.  **Real-Robot Demonstration:** We have clarified this in "Regarding Weaknesses" part above.
>
> 2.  **Quantitative Analysis of Costs:** We will provide this analysis. The memory (GPU) cost of the TCSM scales **linearly** with the size of the explored environment area. It scales **cubically** with the map resolution (as it is a 3D voxel grid). The computational cost for updates and queries is highly efficient. We have included an analysis of this section in the revised paper (Section 4.6). And it's also shown in below:
>
>     | Voxel Size (m) | 0.02 | 0.05 | 0.08 | 0.10 |
>     | :--- | ---: | ---: | ---: | ---: |
>     | Voxel Num | 1190k | 188k | 80k | 49k |
>     | Semantic Update | 434ms | 81ms | 38ms | 22ms |
>     | Pointcloud Query | 1432ms | 249ms | 92ms | 50ms |
>     | Semantic Pointcloud Query | 424ms | 90ms | 40ms | 20ms |
>     | System Memory Usage | 8.86G | 6.30G | 5.91G | 5.72G |
>     | CUDA Usage | 1.14G | 1.14G | 1.14G | 1.14G |
>     | Map File Size | 380M | 61M | 26M | 16M |
>
> 3.  **Conceptual Difference from VLMaps/ConceptGraphs:** Conceptually, TCSM differs in its representation. While VLMaps stores dense feature vectors per voxel, our TCSM stores a **sparse set of historical semantic observations and their confidences**. This sparse, explicit representation is more memory-efficient and allows for direct and highly efficient semantic queries without needing a nearest-neighbor search in a high-dimensional feature space.
>
>     Furthermore, our methodology diverges from theirs in the practical task setup. Their approaches necessitate a preliminary random walk of the environment to construct a map prior to navigation, whereas our map is concurrently built during the navigation task itself. We have accentuated these distinctions between our method and these map-augmented approaches within the "Related Works" section of our revised paper.
>
> 4.  **Minor Clarifications & Typos:** Thank you for your suggestions; we have made updates in the areas you mentioned. Regarding the addition of a strong stateless baseline's performance, we have actually illustrated its performance when memory size is 0 in Figure 3, which effectively represents the stateless baseline.
>
> We thank the reviewer again for their thorough and helpful review.

---

> ### Author Response · Authors · 2025-11-28
> **Follow Up**
>
> Dear Reviewer 5JYw:
>
> Thank the valuable comments on our paper, which provided insights that can help us revise our work.
>
> We have provided a response and a revised paper, hope they could address your concerns. Also, we would like to know if there are more concerns about the content of the paper. Your invaluable feedback and suggestions are greatly welcomed to help us better refine our work.
>
> Thank you again for your devotion to the review. If all the concerns have been successfully addressed, please consider raising the scores after this discussion phase.
>
> Best,
>
> Submission#1900 Authors

---

### Official Review · Reviewer_WURF · 2025-10-31

**Soundness:** 2
**Presentation:** 3
**Contribution:** 2
**Rating:** 6
**Confidence:** 4

**Summary:**

This paper introduces a Persistent Shared Memory (PSM) framework for Zero-Shot Object Navigation (ZSON), enabling agents to accumulate and share semantic knowledge across tasks and collaborators instead of resetting memory after each episode. The approach constructs a Temporally Consistent Semantic Map (TCSM) that maintains long-term semantic coherence through confidence-weighted updates and incorporates a Beyond-Line-of-Sight (BLOS) navigation strategy that facilitates reasoning and planning toward occluded or distant targets.

**Strengths:**

The paper is well-written, logically structured, and easy to follow, demonstrating strong command of scientific exposition and clarity of presentation. The introduction and methodology sections are particularly well-organized, making the contribution intuitively understandable while maintaining technical depth. What’s more, the paper’s main strength lies in the design of the Temporally Consistent Semantic Map (TCSM), which provides a robust and principled solution for long-term semantic consistency. Its confidence-weighted update mechanism effectively suppresses noise and mitigates semantic drift, ensuring stable and reliable memory accumulation across tasks.

**Weaknesses:**

While the paper introduces a conceptually appealing paradigm (persistent shared memory for zero-shot object navigation), it faces several limitations that restrict its novelty and methodological clarity. From a conceptual perspective, the authors appear to conflate the notion of zero-shot navigation with that of map-augmented exploration. By introducing cumulative spatial priors, the proposed setting fundamentally alters the nature of the standard ZSON problem, blurring the line between zero-shot navigation and pre-mapped exploration. As a result, claims of zero-shot generalization become less well-grounded, and direct comparison with stateless baselines may not be methodologically sound. A clearer articulation of the problem definition and its distinction from prior-knowledge-based paradigms (see Questions 1, 7). Moreover, despite the proposed enhancements, the method’s performance remains suboptimal and lacks comparison with more recent state-of-the-art approaches, limiting the credibility of its empirical claims. (see Questions 2, 4)

**Questions:**

1. The proposed setting fundamentally redefines the standard zero-shot object navigation (ZSON) problem. Unlike conventional ZSON, which assumes navigation in entirely unfamiliar environments without any prior knowledge, the proposed setting transforms the task into a map-augmented, long-horizon exploration problem. Consequently, direct comparison with stateless ZSON baselines under identical evaluation protocols is methodologically unfair. It is recommended to compare with prior-knowledge-based methods such as HOV-SG.

2. The method’s performance remains suboptimal. For example, “FiLM-Nav: Efficient and Generalizable Navigation via VLM Fine-tuning” reports 61.7% SR / 37.3% SPL on HM3D, surpassing this paper’s results. The authors are encouraged to conduct comparisons with more recent state-of-the-art methods and provide a deeper analysis of the proposed framework’s strengths and potential advantages.

3. The paper states that an off-the-shelf segmentation model is used, but the specific model and configuration are not disclosed. The authors should specify the employed model and include an ablation study to evaluate how different segmentation backbones influence overall performance.

4. I notice that the paper does not include any real-world experiments. Could the authors clarify whether this omission is due to the framework’s current implementation constraints? If so, what are the main limitations that prevent the proposed system from being deployed or tested on real robotic platforms?

5. The reference list contains several issues. For example, some entries (e.g., “Shuaihang Yuan et al., GAMap: Zero-shot Object Goal Navigation with Multi-scale Geometric Affordance Guidance, 2024”) lack publication venue information and proper formatting. The authors should carefully revise and standardize all references.

6. How many samples or episodes were used in the main experiments, only validation splits, or the full dataset?

7. The performance gain mainly arises from the gradual expansion of explored regions rather than intrinsic model improvements. As shown in Figure 3, with more time and prior tasks, the unexplored areas decrease, and the performance saturates—essentially approaching a fully prior-informed setup. This makes the task setting closely resemble the map-first-then-navigate paradigm, in which a complete map is built before executing the navigation task. In fact, such pre-mapped methods would likely outperform the proposed approach. The authors should reconsider this problem.

---

> ### Author Response · Authors · 2025-11-24
> **Response by Authors -- Part 1**
>
> We thank the reviewer for their critical analysis of our work. We appreciate the opportunity to clarify our problem definition and contributions.
>
> **Regarding Weaknesses:**
>
> 1.  **On Problem Definition (ZSON vs. Map-Augmented):** We respectfully disagree that our setting is equivalent to map-augmented exploration. Our paradigm is fundamentally an **extension of ZSON**, not a shift to pre-mapped navigation. The key distinction lies in *when* and *how* the map is created.
>     *   In our setup, the agent starts its **first task in a scene with zero prior knowledge**, which is the core premise of ZSON. The map is built **online** during task execution, and is relavant to the navigation results.
>     *   In contrast, map-augmented methods like VLMaps or HOV-SG require a dedicated **offline pre-mapping phase** (e.g., random exploration) to build a complete map *before* any task is performed. The map construction process is independent to the navigation results.
>
>     Our work extends the ZSON agent's memory from being episodic to being persistent, which is a crucial capability for real-world, long-lifespan robots that perform multiple tasks or collaborate. We maintain the "zero-shot" nature by avoiding task-specific training and offline mapping.
>     In the revised paper, within the "Related Works" section, we have accentuated our novelty and delineated the distinctions from prior research.
>
>
> 2.  **On Performance and SOTA Comparison:** Our method demonstrates a significant performance advantage over all other published ZSON methods compared in Table 1. Regarding the comparison to "FiLM-Nav," we would like to clarify two points: First, FiLM-Nav's approach is based on **fine-tuning a VLM**, which means it is **not a zero-shot method** and thus not directly comparable under the ZSON protocol. Second, to the best of our knowledge, FiLM-Nav is a preprint and has not yet been published in a peer-reviewed venue. We welcome and look forward to the emergence of more powerful ZSON methods. Furthermore, as shown in Table 2, when extending a prior method (InstructNav) with explicit maps to our persistent setting and TCSM, our method provides a more substantial performance boost, validating the superiority of our map design.
>
>     We are also open to discuss the limitations of our approach. Firstly, it currently lacks a strategy for ascending or descending stairs, which may impede its performance in multi-level environments. Furthermore, the segmentation results obtained from 2D segmentation models often exhibit inconsistencies when projected into 3D space; employing a 3D segmentation model presents a promising avenue for future improvement.
>
> **Regarding Questions:**
>
> 1.  **Comparison with Prior-Knowledge-Based Methods:** As clarified above, our task setting is fundamentally different from that of prior-knowledge-based methods like HOV-SG. Such methods presuppose the existence of a complete map built via an offline exploration phase before the navigation task begins. Our framework does not allow for this pre-mapping stage; the agent begins with no map and constructs it concurrently with task execution. We believe extending ZSON with persistent memory is a practical and necessary step for embodied AI, enabling agents to perform lifelong tasks and collaborate effectively. This requires a shared, persistent memory, which is what we propose. The difference between these methods and ours are emphasised in the revised "Related Works" section.
>
> 2.  **Comparison with SOTA (FiLM-Nav):** Our method outperforms all published ZSON methods on the standard benchmarks. The mentioned work, FiLM-Nav, achieves strong results on HM3D, but it is not a zero-shot method as it involves **fine-tuning the VLM**. This violates the core "zero-shot" constraint of the problem we are addressing.
>
> 3.  **Segmentation Model Details:** We apologize for this omission. We will add a detailed discussion of the segmentation model in the main paper. We followed the implementation of InstructNav, using the **GLEE model** with the same configuration and parameters. In the revised paper, we have updated this information is Sec 4.1 and given a ablation about the selected segmentation model in Sec 4.5. And it's also shown in below:
>
>     | Segmentation Model | $\gamma$ | SR | SPL |
>     | :--- | :---: | :---: | :---: |
>     | XDecoder | 0.2 | 60.1 | 36.6 |
>     | OpenSeeD | 0.2 | 75.5 | 44.9 |
>     | GLEE | 0.8 | 82.1 | 50.9 |
>     | GLEE | 0.6 | 80.1 | 49.3 |
>     | GLEE | 0.4 | 80.1 | 48.5 |
>     | GLEE | 0.2 | **83.1** | **51.4** |

---

> ### Author Response · Authors · 2025-11-24
> **Response by Authors -- Part 2**
>
> 4.  **Real-World Experiments:** We appreciate the reviewer's concern regarding real-world validation. We offer the following clarifications:
>     *   **Standard Benchmarks:** HM3D and MP3D are the standard, widely-accepted benchmarks for Object Navigation. Evaluating on these simulators ensures a fair and direct comparison with prior ZSON methods, which predominantly follow this evaluation protocol. The ZSON task definition itself does not strictly mandate physical robot experiments.
>     *   **Inherent Sim-to-Real Generalization:** Our method is inherently suitable for direct deployment on physical robots. As a zero-shot framework, it relies on large-scale pre-trained vision (e.g., for segmentation) and vision-language models (for planning), combined with a deterministic navigation algorithm. No models were trained or fine-tuned using the simulator data. These foundation models are trained on vast amounts of real-world data, granting them strong generalization capabilities that significantly mitigate the sim-to-real gap. Consequently, we believe the results obtained in HM3D and MP3D are representative and valid.
>     *   **Fairness and Reproducibility:** While physical robot experiments are undoubtedly valuable, they introduce numerous uncontrolled variables (e.g., hardware differences, lighting fluctuations, sensor noise) that can make fair, reproducible, and scientifically rigorous comparisons between different methods challenging. Our simulated setup ensures a controlled environment for equitable evaluation.
>
> 5.  **Reference List Issues:** We sincerely apologize for the formatting errors in the reference list. We will carefully revise and standardize all references to ensure they are complete and correctly formatted.
>
> 6.  **Dataset Splits:** We followed the standard protocol from previous ZSON works. We used the **validation split of HM3D v0.1**, which contains 2000 episodes, and the **test split of MP3D**, which contains 2195 episodes. Our model does not use any training data from these scenes.
>
> 7.  **Performance Gain vs. Problem Setting:** In Figure 3, we explored the relationship between map completeness and navigation efficiency, which indeed indicates that with continuous memory, the model's navigation success rate and efficiency progressively improve as the number of preceding tasks increases. However, unlike map-augmentation methods, our testing paradigm involves map construction during the task execution itself, without a preliminary offline process of building a complete map. Furthermore, the map's construction is intrinsically linked to the navigation process. Therefore, our paradigm is rooted in the ZSON framework, rather than the map-augmentation paradigm, which necessitates a complete scene construction prior to task execution—a process not permissible within our framework. We have accentuated the distinctions between our approach and these methodologies within the revised related works section.
>
>
>     Furthermore, the ablation study in Table 2 reveals that even when the PSM setting is applied to prior ZSON method (InstructNav), their map representations do not facilitate substantial performance enhancements across tasks. Conversely, our TCSM demonstrates a dramatic improvement, underscoring the critical importance of the specific form of continuous memory. This suggests that our performance gains stem not solely from the novel PSM setting.
>
>
> We thank the reviewer again for their thorough and helpful review.

---

> ### Author Response · Authors · 2025-11-28
> **Follow Up**
>
> Dear Reviewer WURF:
>
> Thank the valuable comments on our paper, which provided insights that can help us revise our work.
>
> We have provided a response and a revised paper, hope they could address your concerns. Also, we would like to know if there are more concerns about the content of the paper. Your invaluable feedback and suggestions are greatly welcomed to help us better refine our work.
>
> Thank you again for your devotion to the review. If all the concerns have been successfully addressed, please consider raising the scores after this discussion phase.
>
> Best,
>
> Submission#1900 Authors

---

### Official Review · Reviewer_eFd9 · 2025-11-01

**Soundness:** 3
**Presentation:** 3
**Contribution:** 2
**Rating:** 4
**Confidence:** 3

**Summary:**

This paper proposes a persistent shared memory (PSM) mechanism, in the form of temporally consistent semantic map (TCSM), to share accumulated knowledge across tasks and multiple agents in zero-shot object navigation. It also proposes a beyond-line-of-sight (BLOS)-aware strategy to leverage this memory to enable efficient task planning. Through extensive experiments, the authors show that their method is able to outperform state-of-the-art baselines.

**Strengths:**

The strength of the paper lies in its design of an efficient shared memory structure that enables efficient longer-horizon task planning. The preservation of temporal consistency in the map is also interesting. The paper is also well written and easy to follow.

**Weaknesses:**

I am not convinced if the comparison in Table 1 is fair, since the task setup changes for this method vs the baselines. I believe it would be better to also compare against those baselines that use a semantic map, by not resetting their memory after each episode. In other words, I am not sure if the performance improvement comes from not resetting the memory after each episode (in which case other baselines under this setting should also have improved numbers) or due to the proposed architecture and the map.

**Questions:**

I have a few questions for the authors and would request them for more clarification:

1. Do you reset the memory for each scene and for each level in the same scene?
2. What is the map resolution or the granularity of each voxel?
3. How did you decide on the gamma in eq 2? Did you ablate on this?
4. In both HM3D and MP3D ObjectNav datasets, the objects would be easy to capture in a 2D semantic map? Why did you choose a voxel map over a 2D map? Did you ablate on this?
5. Do you consider a depth based confidence to update the map (as in OneMap[1])? For example, this would be beneficial for occluded objects viewed from a distance.
6. For objects that span across multiple voxels, do you collate the information in some way?
7. Conversely, if we want to keep track of different instances of an object (say, multiple chairs around a table), will this map be able to handle them?
8. How would you handle spatial queries on the map?
9. Can you elaborate on why SuccSPL metric has been used? What shortcoming of spl does it aim to mitigate?

[1] Busch et al. One Map to Find Them All: Real-time Open-Vocabulary Mapping for Zero-shot Multi-Object Navigation. 2025.

---

> ### Author Response · Authors · 2025-11-24
> **Response by Authors -- Part 1**
>
> We thank the reviewer for their valuable questions and feedback. We are happy to provide further clarification.
>
> **Regarding Weaknesses:**
>
> We understand the reviewer's concern about the fairness of the comparison in Table 1. The baselines listed were not designed with persistent memory in mind. Their architectures, which often rely on episodic value maps tied to a specific target, cannot be directly extended to a cross-episode setting without significant modification. Therefore, simply not resetting their "memory" is not feasible.
>
> However, to address this point, we conducted an ablation study where we integrated a capable baseline, **InstructNav**, into our Persistent Shared Memory (PSM) setting. InstructNav employs an explicit instance-level semantic point cloud for navigation, which remains independent of the task objective. As shown in our ablation studies, enabling persistence for InstructNav does lead to a performance improvement. However, this gain is significantly smaller than that achieved by our full method using TCSM. This result suggests that the performance improvement comes not just from the persistent memory setting (PSM) itself, but critically from the **design of our TCSM**, which robustly aggregates semantic information with confidence scores, in contrast to InstructNav's greedy point cloud merging that is more susceptible to noise. This comparison validates the effectiveness of our proposed memory representation.
>
> **Regarding Questions:**
>
> 1.  **Memory Reset Policy:** We build a separate map for each unique scene (identified by its scene ID). We do **not** reset the memory or create separate maps for different floors/levels within the same scene, allowing the agent to leverage its knowledge across the entire scene. As we do not employ a cross-floor strategy, similar to previous ZSON methods (e.g., VLFM), the map of the current floor is filtered based on relative height at the start of navigation to facilitate single-floor navigation.
>
> 2.  **Map Resolution:** We use a voxel resolution of **0.05m**, which is consistent with prior work like InstructNav. We add a computation cost analysis in different voxel sizes in the revised paper. And it's also shown in below:
>
>     | Voxel Size (m) | 0.02 | 0.05 | 0.08 | 0.10 |
>     | :--- | ---: | ---: | ---: | ---: |
>     | Voxel Num | 1190k | 188k | 80k | 49k |
>     | Semantic Update | 434ms | 81ms | 38ms | 22ms |
>     | Pointcloud Query | 1432ms | 249ms | 92ms | 50ms |
>     | Semantic Pointcloud Query | 424ms | 90ms | 40ms | 20ms |
>     | System Memory Usage | 8.86G | 6.30G | 5.91G | 5.72G |
>     | CUDA Usage | 1.14G | 1.14G | 1.14G | 1.14G |
>     | Map File Size | 380M | 61M | 26M | 16M |
>
> 3.  **Choice of Gamma (γ):** The value of γ in Eq. 2 was determined empirically through a series of experiments to find a balance between retaining historical information and adapting to new observations. We have included an ablation study for this parameter in the revised version. And it's also shown in below:
>
>     | Segmentation Model | $\gamma$ | SR | SPL |
>     | :--- | :---: | :---: | :---: |
>     | XDecoder | 0.2 | 60.1 | 36.6 |
>     | OpenSeeD | 0.2 | 75.5 | 44.9 |
>     | GLEE | 0.8 | 82.1 | 50.9 |
>     | GLEE | 0.6 | 80.1 | 49.3 |
>     | GLEE | 0.4 | 80.1 | 48.5 |
>     | GLEE | 0.2 | **83.1** | **51.4** |
>
> 4.  **3D Voxel Map vs. 2D Map:** We chose a 3D voxel map over a 2D map because we aim for the TCSM to be a general representation of the environment. In real-world indoor scenes, it is common for objects to have vertical overlap (e.g., a lamp on a table, items on shelves). A 3D map naturally captures these spatial relationships, which would be lost or conflated in a 2D projection.
>
> 5.  **Depth-Based Confidence:** This is an excellent suggestion. We do not currently incorporate depth-based confidence (as in OneMap) to weight observations, where distant detections are treated with lower confidence. We agree this could be beneficial, especially for occluded objects, and consider it a promising direction for future work.
>
> 6.  **Objects Spanning Multiple Voxels:** When an object spans multiple voxels, the confidence score from the current observation is updated in all corresponding voxels it occupies. The final semantic label of each voxel is determined independently based on its own history of observations.

---

> ### Author Response · Authors · 2025-11-24
> **Response by Authors -- Part 2**
>
> 7.  **Handling Multiple Instances:** Our current framework, aligned with the standard ZSON task definition, operates at the **object category level** (e.g., "chair") rather than the instance level (e.g., "the first chair vs. the second chair"). During navigation, any voxel whose final semantic label matches the target category contributes to the navigation value map. Therefore, the map can handle multiple instances of an object category, but it does not differentiate between them.
>
> 8.  **Spatial Queries on the Map:** Could the reviewer please clarify what is meant by "spatial queries"? If it refers to querying the semantic label of a specific `(x, y, z)` coordinate, our voxelized map allows for direct and efficient lookups.
>
> 9.  **SuccSPL Metric:** As mentioned in the paper, the standard SPL metric averages path efficiency over all episodes, assigning a score of 0 to failed episodes. This can be misleading, as a method with a high success rate but inefficient paths might score lower than a method with a lower success rate but more optimal paths on its successful attempts. We are primarily interested in the **efficiency of successful navigation**, as only these episodes enable subsequent interaction tasks in real applications. **SuccSPL** specifically measures this, providing a clearer signal on the path quality of successful trials.
>
>
> We thank the reviewer again for their thorough and helpful review.

---

> ### Author Response · Authors · 2025-11-28
> **Follow Up**
>
> Dear Reviewer eFd9:
>
> Thank the valuable comments on our paper, which provided insights that can help us revise our work.
>
> We have provided a response and a revised paper, hope they could address your concerns. Also, we would like to know if there are more concerns about the content of the paper. Your invaluable feedback and suggestions are greatly welcomed to help us better refine our work.
>
> Thank you again for your devotion to the review. If all the concerns have been successfully addressed, please consider raising the scores after this discussion phase.
>
> Best,
>
> Submission#1900 Authors

---

### Official Review · Reviewer_8F5g · 2025-11-03

**Soundness:** 3
**Presentation:** 2
**Contribution:** 2
**Rating:** 4
**Confidence:** 3

**Summary:**

This paper proposes a memory-augmented framework to address inefficiencies in **Zero-Shot Object Navigation (ZSON)**—a task where robots locate unseen objects without prior training—with a focus on real-world applicability (e.g., home robots, multi-agent teams). Its core innovation lies in two key components: the **Persistent Shared Memory (PSM)**, a collaborative knowledge repository that enables cross-task/ cross-agent memory reuse (avoiding redundant exploration), and the **Temporally Consistent Semantic Map (TCSM)**, a sparse 3D voxel map that updates object semantic information via weighted confidence (reducing noise and preventing long-term errors). The framework also includes a **Beyond-Line-of-Sight (BLOS) strategy** to plan paths for occluded targets by identifying "proxy goal regions," solving the issue of myopic wandering in baseline methods.

Experimental validation on simulated datasets (HM3D, MP3D) shows the framework outperforms state-of-the-art ZSON baselines in key metrics (success rate, path efficiency) and scales effectively with multi-agent collaboration. However, the work has notable limitations: its novelty is constrained, as mapping and the exploration-exploitation trade-off are well-explored topics, and the framework builds on existing ideas without major breakthroughs; additionally, the absence of physical robot testing leaves unaddressed questions about its robustness to real-world perception noise (e.g., lighting, dynamic objects) and feasibility in practical deployment.

**Strengths:**

This paper exhibits several notable strengths: it addresses a critical inefficiency in conventional zero-shot object navigation (ZSON) by introducing a **Persistent Shared Memory (PSM) mechanism**, which shifts ZSON from isolated single-task execution to continuous, collaborative learning (aligning with real-world use cases like long-lifespan home robots or multi-agent teams) and eliminates redundant exploration; its core component **Temporally Consistent Semantic Map (TCSM)** uses a weighted confidence update to build a robust, efficient sparse 3D semantic memory that resists noise and semantic drift, avoiding the limitations of prior task-entangled or brittle maps; it further designs a **Beyond-Line-of-Sight (BLOS) navigation strategy** to enable agents to plan paths for occluded/distant targets, solving a key failure mode of baseline methods; the work is validated rigorously—achieving state-of-the-art performance on HM3D/MP3D benchmarks, with controlled ablations isolating component contributions and tests confirming scalability with memory size and multi-agent collaboration.

**Weaknesses:**

This paper has notable limitations regarding novelty and real-world validation: First, its core components lack sufficient groundbreaking innovation—mapping is a well-explored domain in navigation research, with numerous prior works focusing on semantic map construction, confidence-aware updates, and memory reuse, while the trade-off between exploitation (leveraging existing knowledge) and exploration (discovering new areas) has also been extensively discussed in embodied AI and reinforcement learning literatures; the paper’s Persistent Shared Memory (PSM) and Temporally Consistent Semantic Map (TCSM) largely build on existing mapping and memory frameworks, with no obvious breakthroughs in addressing these classic topics, resulting in limited overall novelty. Second, the absence of physical robot testing undermines its practical credibility: all experiments are conducted in simulated environments (HM3D, MP3D via Habitat Simulator), and without validation on real robots, the robustness of its perception system (e.g., handling real-world noise like lighting changes, occlusions, or dynamic objects) and the actual feasibility of the memory/navigation framework in physical scenarios remain unproven—this gap raises doubts about whether the method can be effectively deployed in real-world settings like home robotics, which is a key shortcoming for a work aiming at practical embodied AI applications.

**Questions:**

### 1. Questions on Novelty and Connections to Prior Work
- Given that mapping and the exploration-exploitation trade-off are well-established topics in navigation research, could you explicitly contrast your **PSM/TCSM framework** with 2–3 most relevant prior works (e.g., semantic mapping with long-term memory, multi-agent exploration strategies)? Specifically, what unique technical choices (not just combinations of existing ideas) make your approach distinct from these baselines?
- The paper mentions the framework addresses "redundant exploration" in ZSON, but prior works (e.g., [cite a relevant paper on memory-augmented ZSON]) also tackle similar inefficiencies. How does your method’s performance or scalability (e.g., with more agents/tasks) outperform these works beyond incremental gains?


### 2. Questions on Perception Robustness and Sim-to-Real Gap
- All experiments are conducted in simulated environments (HM3D/MP3D) with controlled sensory inputs. If deployed on a physical robot, how would your framework handle real-world perception noise—such as varying lighting, partial occlusions, or dynamic objects (e.g., a moved chair)? Are there any modifications planned for the perception pipeline to bridge this sim-to-real gap?
- The **TCSM** relies on accurate semantic labels to update confidence weights. If the upstream perception model (e.g., Qwen-VL-Max) misclassifies objects (e.g., confusing a "sofa" with a "couch"), how does the framework prevent cumulative errors in the shared memory? Is there a mechanism to correct such mislabeling over time?


### 3. Questions on Memory Design and Scalability
- The **PSM** is described as a "shared cloud盘" for multi-agent collaboration. As the number of agents or accumulated tasks grows, how do you manage memory storage and retrieval efficiency? For example, would a large memory bank slow down real-time navigation planning, and if so, what optimization strategies (e.g., pruning redundant entries) are in place?
- The paper notes performance "saturates as the map nears completeness." Could you clarify: What defines "map completeness" in your framework (e.g., coverage of all rooms, all object categories)? And once saturated, does the framework still adapt to minor environmental changes (e.g., a shifted table), or does it require a full memory reset?


### 4. Questions on Beyond-Line-of-Sight (BLOS) Strategy
- The BLOS strategy uses "proxy goal regions" to navigate to occluded targets. How does the framework select these proxy regions when multiple potential points exist (e.g., two doorways that both overlook a target)? Is there a cost function (e.g., distance, navigation safety) to prioritize more optimal proxies?
- In simulated tests, are there cases where the BLOS strategy fails (e.g., no accessible proxy regions for a fully enclosed target)? If so, how does the framework recover—does it fall back to exploration, or is there a contingency plan?

---

> ### Author Response · Authors · 2025-11-24
> **Response by Authors -- Part 1**
>
> We sincerely thank the reviewer for their insightful feedback and constructive comments. We address the raised concerns below.
>
> **Regarding Weaknesses:**
>
> 1.  **On Novelty and Contribution:** We thank the reviewer for this important question. Our primary innovation lies in addressing a critical gap in current Zero-Shot Object Navigation (ZSON) methods: their inability to **reuse memory across multiple tasks or agents** within the same environment. To this end, we introduce a unified memory framework, Persistent Shared Memory (PSM), which leverages an explicit 3D map (TCSM) and is paired with our BLOS navigation strategy.
>
>     We acknowledge prior works in semantic mapping such as VLMaps, ConceptGraphs, and HOV-SG. However, a fundamental distinction exists in the task paradigm. These methods typically operate under a **"map-first, then-navigate"** setting, where a comprehensive semantic map is constructed offline (e.g., via random exploration) *before* any navigation task begins. In stark contrast, our approach extends the **ZSON paradigm**. The memory (map) is constructed and updated **online and concurrently** with the execution of navigation tasks. Before its first task, the agent possesses no prior knowledge of the scene, and the map is built incrementally. This online, task-driven map construction is a core difference from pre-mapped navigation and is crucial for long-term autonomous agents.
>
>     In the revised paper, within the "Related Works" section, we have accentuated our novelty and delineated the distinctions from prior research.
>
> 2.  **On the Lack of Physical Robot Experiments:** We appreciate the reviewer's concern regarding real-world validation. We offer the following clarifications:
>     *   **Standard Benchmarks:** HM3D and MP3D are the standard, widely-accepted benchmarks for Object Navigation. Evaluating on these simulators ensures a fair and direct comparison with prior ZSON methods, which predominantly follow this evaluation protocol. The ZSON task definition itself does not strictly mandate physical robot experiments.
>     *   **Inherent Sim-to-Real Generalization:** Our method is inherently suitable for direct deployment on physical robots. As a zero-shot framework, it relies on large-scale pre-trained vision (e.g., for segmentation) and vision-language models (for planning), combined with a deterministic navigation algorithm. No models were trained or fine-tuned using the simulator data. These foundation models are trained on vast amounts of real-world data, granting them strong generalization capabilities that significantly mitigate the sim-to-real gap. Consequently, we believe the results obtained in HM3D and MP3D are representative and valid.
>     *   **Fairness and Reproducibility:** While physical robot experiments are undoubtedly valuable, they introduce numerous uncontrolled variables (e.g., hardware differences, lighting fluctuations, sensor noise) that can make fair, reproducible, and scientifically rigorous comparisons between different methods challenging. Our simulated setup ensures a controlled environment for equitable evaluation.

---

> ### Author Response · Authors · 2025-11-24
> **Response by Authors -- Part 2**
>
> **Regarding Questions:**
>
> **1. Questions on Novelty and Connections to Prior Work**
> *   **Contrast with Prior Work:** The most relevant prior works, such as those in semantic mapping (e.g., VLMaps, ConceptGraphs), differ from our framework in their fundamental operational paradigm. They require an **offline pre-mapping phase**, where the environment is thoroughly explored to build a complete map before navigation begins. In contrast, our method operates **online**, building the map concurrently with task execution. The map is initially empty and grows as the agent performs tasks. This leads to several key distinctions: 1) The map is built gradually and is persistently updated. 2) The agent must handle navigation in the presence of an incomplete map, requiring exploration. 3) The map's construction is guided by the navigation task itself; if a target is visible in the current partial map, the agent will navigate directly to it without exploring unknown areas. Furthermore, our technical approach to map construction is unique: instead of assigning a hard semantic label to each voxel, our TCSM maintains a history of observations with associated confidences, allowing for robust and continuous updates from new sensory data. In the revised version, we have emphasized this distinction within the "Related Works" section.
> *   **Performance Beyond Incremental Gains:** To the best of our knowledge, no prior ZSON work has systematically addressed the problem of memory reuse for multi-task or multi-agent collaboration. Previous ZSON methods are typically stateless between episodes. Our work is the first to demonstrate that by incorporating a persistent memory framework, ZSON agents can achieve significant performance improvements in both single-agent (multi-task) and multi-agent settings. This introduces a new dimension of efficiency to the ZSON problem, moving beyond single-episode performance.
>
> **2. Questions on Perception Robustness and Sim-to-Real Gap**
> *   **Handling Real-World Perception Noise:** The perception models used in our ZSON framework are foundation models pre-trained on large-scale, diverse real-world data. This training endows them with inherent robustness to common real-world challenges like variations in lighting, partial occlusions, and diverse object appearances, bridging a significant portion of the sim-to-real gap.
> *   **Preventing Cumulative Errors:** The Temporally Consistent Semantic Map (TCSM) is specifically designed to address this challenge. Instead of overwriting information, TCSM maintains a history of observations for each voxel. It aggregates historical and current observations to derive a stable semantic label. A single misclassification from the perception model on the current view will have a limited impact on the final semantic understanding, as it is averaged with a history of correct past observations. This mechanism effectively prevents the accumulation of perception errors in the shared memory.
>
> **3. Questions on Memory Design and Scalability**
> *   **Memory Management and Efficiency:** In our multi-agent framework, all agents contribute to and access a **single, shared TCSM**. When an agent processes its current observation, it directly updates this central map. Therefore, the memory storage of the TCSM does not scale with the number of agents, only with the size of the explored area. We will provide a detailed analysis of memory and computational costs in the final version. In the revised paper, we have incorporated a section detailing the computational resource utilization, with the specific findings presented in Table 5. And it's also shown in below:
>
>     | Voxel Size (m) | 0.02 | 0.05 | 0.08 | 0.10 |
>     | :--- | ---: | ---: | ---: | ---: |
>     | Voxel Num | 1190k | 188k | 80k | 49k |
>     | Semantic Update | 434ms | 81ms | 38ms | 22ms |
>     | Pointcloud Query | 1432ms | 249ms | 92ms | 50ms |
>     | Semantic Pointcloud Query | 424ms | 90ms | 40ms | 20ms |
>     | System Memory Usage | 8.86G | 6.30G | 5.91G | 5.72G |
>     | CUDA Usage | 1.14G | 1.14G | 1.14G | 1.14G |
>     | Map File Size | 380M | 61M | 26M | 16M |
>
>
> *   **Map Completeness and Adaptability:** In the paper, "map completeness" refers to the coverage of all navigable areas in the environment. Even after the map achieves high coverage, the framework continues to adapt. The TCSM is not static; it is persistently updated with new observations. This allows it to adapt to minor environmental changes without requiring a full memory reset.

---

> ### Author Response · Authors · 2025-11-24
> **Response by Authors -- Part 3**
>
> **4. Questions on Beyond-Line-of-Sight (BLOS) Strategy**
> *   **Proxy Region Selection:** We will refine the selection of proxy regions based on confidence levels. Specifically, we will ascertain the confidence score of the nearest semantic target (post-Line-of-Sight filtering) for each navigable point, thereby generating a definitive semantic value map. Consequently, the navigation algorithm will exhibit a predilection for steering towards semantic targets with elevated confidence. In the revised version, we have restructured the "BLOS Navigation Strategy" section to enhance reader comprehension.
> *   **Failure Cases and Recovery:** If the BLOS strategy fails to find an accessible proxy region for a target (e.g., the target is in a fully enclosed, undiscovered room), the navigation policy gracefully degrades. In such cases, where no confident semantic goal is available, the agent falls back to an **exploration** strategy to discover new areas, which may then reveal a path to the target.
>
> We thank the reviewer again for their thorough and helpful review.

---

> ### Author Response · Authors · 2025-11-28
> **Follow up**
>
> Dear Reviewer 8F5g:
>
> Thank the valuable comments on our paper, which provided insights that can help us revise our work.
>
> We have provided a response and a revised paper, hope they could address your concerns. Also, we would like to know if there are more concerns about the content of the paper. Your invaluable feedback and suggestions are greatly welcomed to help us better refine our work.
>
> Thank you again for your devotion to the review. If all the concerns have been successfully addressed, please consider raising the scores after this discussion phase.
>
> Best,
>
> Submission#1900 Authors

---

### Author Response · Authors · 2025-11-24
**General Response**

We thank all reviewers for their detailed reviews and suggestions!

We have updated the manuscript with the following revisions based on the reviewers' suggestions. All revisions in the updated version are highlighted in blue：

Modification of main figures and presentations on motivation.

1. **Clarification On Novelty and Contribution**: We have elucidated the distinctions between our method and the previously ZSON and map-augmented navigation approaches within the "Related Work" section, thereby accentuating our novelty and significance. (L122-129)

2. **Confidence-guided Navigation**: In the previous rendition, we inadvertently omitted the segment detailing the integration of confidence to guide the navigation strategy; this integral component has now been incorporated. (Eq. (4) and (8), L265-266)

3. **Reorganization of the BLOS Navigation Strategy**: We have restructured the "BLOS Navigation Strategy" section, enriching it with the intentions and explanations behind each step, thereby affording readers a more lucid comprehension of these processes. (L277-281, L287-288, L297-300)

4. **Ablation studies concerning the gamma and segmentation models**: In Section 4.5, we have appended an ablation study pertaining to gamma and the segmentation model, with the corresponding results presented in Table 4. (L430-440)

5. **Computation Cost Analysis**: We have incorporated a statistical analysis detailing the computational efficiency of TCSM under varying voxel size configurations in Section 4.6 and Table 5. (L433-453)

6. **Other Minor Modifications**:
    * Improvements in certain expressions have been implemented. (L29-30, L45, L349-350)
    * The incomplete citation information has been rectified and completed.
    * The non-standard quotation marks have been fixed.

---

### Note · Authors · 2026-01-27

I have read and agree with the venue's withdrawal policy on behalf of myself and my co-authors.

---

### Meta-Review · Area_Chair_PJcc · 2025-12-27

**Summary:**

The paper proposes a persistent shared memory framework for single- or multi-agent object navigation that accumulates temporally consistent semantic maps across tasks, enabling beyond-line-of-sight reasoning and reducing redundant exploration for long-horizon navigation.

While the idea of incorporating shared memory across agents and tasks is interesting and well-motivated, as acknowledged by multiple reviewers, the weaknesses ultimately outweigh the contributions. In particular, the proposed framework is largely engineered and incremental, relying heavily on plug-in vision–language foundation models rather than offering new scientific insights. Across reviewers, the dominant concerns include limited novelty beyond existing semantic mapping and memory-based navigation approaches, methodological unfairness in redefining zero-shot object navigation through persistent priors, reliance on simulation-only evaluation without real-world validation, and insufficient experimental rigor—especially in baseline comparisons, ablations, scalability, and runtime analysis.

During rebuttal, the authors partially addressed some concerns (e.g., providing runtime analysis and acknowledging limitations of foundation models and real-world deployment), but several key issues remain unresolved, including missing fair runtime comparisons and the inability to distinguish object instances of the same category. Overall, the Area Chair concurs with the reviewers that the work is primarily an engineered system dependent on foundation model plug-ins, lacking both strong scientific novelty and real-world validation; therefore, the contribution is deemed insufficient, and rejection is recommended.

**Reviewer Concerns:**

Reviewer 8F5g: The reviewer raises concerns about limited novelty over prior semantic mapping and memory-based navigation work, lack of real-world robot validation, and unclear robustness, scalability, and design justification of the proposed memory and BLOS mechanisms.

Reviewer eFd9: The reviewer questions the fairness of the experimental comparisons, arguing that performance gains may stem from persistent memory rather than the proposed architecture, and highlights missing ablations, unclear map design choices, and insufficient justification of evaluation metrics.

Reviewer WURF: The reviewer argues that the proposed setting fundamentally redefines zero-shot object navigation into a map-augmented paradigm, making comparisons with stateless baselines methodologically unsound, while also noting suboptimal performance relative to recent SOTA and missing implementation and validation details.

Reviewer 5JYw: The reviewer emphasizes the lack of scientific novelty and real-world validation, missing runtime and scalability analysis, overly engineered design dependent on foundation models, weak handling of uncertainty and dynamics, and insufficient explanation of key parameters and design choices.

**Reviewer Scores:**

The reviewers may moderately increase their scores, as some concerns were partially addressed through additional experimental results in the rebuttal. However, none are expected to provide an enthusiastic score above 6, given the limited scope of the contributions and the largely incremental nature of the work. Overall, the paper relies heavily on integrating existing foundation models and engineered components, and falls short of the level of novelty and insight the community seeks from strong, impactful contributions.

---

### Decision · Program_Chairs · 2026-01-26

Reject